# Telework and Worker Health and Well-Being: A Review and Recommendations for Research and Practice

**DOI:** 10.3390/ijerph19073879

**Published:** 2022-03-24

**Authors:** Julia L. O. Beckel, Gwenith G. Fisher

**Affiliations:** 1Department of Psychology, Colorado State University, Fort Collins, CO 80523, USA; gwen.fisher@colostate.edu; 2Department of Environmental and Occupational Health, Department of Epidemiology, Colorado School of Public Health, Aurora, CO 80045, USA

**Keywords:** telework, remote work, flexible work arrangements, worker health, worker well-being

## Abstract

Telework (also referred to as telecommuting or remote work), is defined as working outside of the conventional office setting, such as within one’s home or in a remote office location, often using a form of information communication technology to communicate with others (supervisors, coworkers, subordinates, customers, etc.) and to perform work tasks. Remote work increased over the last decade and tremendously in response to the COVID-19 pandemic. The purpose of this article is to review and critically evaluate the existing research about telework and worker health and well-being. In addition, we review and evaluate how engaging in this flexible form of work impacts worker health and well-being. Specifically, we performed a literature search on the empirical literature related to teleworking and worker health and well-being, and reviewed articles published after the year 2000 based on the extent to which they had been discussed in prior reviews. Next, we developed a conceptual framework based on our review of the empirical literature. Our model explains the process by which telework may affect worker health and well-being in reference to individual, work/life/family, organizational, and macro level factors. These components are explained in depth, followed by methodological and fundamental recommendations intended to guide future research, policies, and practices to maximize the benefits and minimize the harms associated with telework, and offer recommendations for future research.

## 1. Introduction

Telework refers to working outside of the office or another physical organizational setting, such as within one’s home or from another location, often using a form of information communication technology to perform work tasks and communicate with others both in and outside the organization [1]. To date, various organizational, political, and social factors have contributed to the rise and development of telework programs in the United States (U.S.) For a review of the history of telework, see Allen et al. [2] Research about the effectiveness of telework has gained popularity within the past decade and telework has recently emerged as a highly important and relevant issue due to its increased prevalence during the COVID-19 pandemic [3]. For instance, at the height of the pandemic approximately 70% of United States (US) workers with jobs conducive to telework were working from home or in a remote capacity [4]. Prior to the pandemic, approximately 3.6% of the U.S. workforce and 5.4% of all workers in the European Union (EU) reported teleworking full-time, and a greater number (approximately 43% of workers in the U.S. and 9% in the EU) reported teleworking from home at least some of the time.

First, it is important to define what we mean by telework to clarify the various terms that have been used to describe this type of work arrangement. Terms such as telecommuting, remote work, homework, virtual work, flexible work, and distributed work have been used interchangeably with alternating definitions in the literature [2]. This lack of consensus has led to challenges when evaluating prior research and findings due to changes in the implementation and location of this form of work. For the purpose of this review, we rely upon the following definition provided by Allen et al. [2]:
“Telecommuting is a work practice that involves members of an organization substituting a portion of their typical work hours (ranging from a few hours a week to nearly full-time) to work away from a central workplace—typically principally from home—using technology to interact with others as needed to conduct work tasks”(p. 44)

The purpose of this article is to summarize research regarding the associations between telework and worker health and well-being based on a thorough and multi-disciplinary review of the telework, work design, ergonomics, and occupational health psychology literature. Prior research on telework has largely focused on work-related outcomes, such as performance, rather than worker health and well-being when considering telework. However, due to the increased prevalence of telework over the last decade and the sudden and large increase due to the COVID-19 pandemic, there is a critical need to understand how teleworking may impact workers’ physical and psychological well-being.

Prior literature reviews have also left substantial gaps relating to our understanding of how telework relates to worker health and well-being. Bailey and Kurland [1] discussed definitional and methodological challenges associated with telework research, as well as demographics relating to the “who, where, and why” of teleworkers. However, their review did not provide comprehensive coverage of the potential outcomes of teleworking. In a more recent review, Allen and colleagues [2] summarized the state of telework research, citing the importance of the extent of telework (i.e., the proportion of one’s time spent teleworking) in research, as well as provide a thorough explanation of outcomes related to both work and social/family outcomes. However, the authors did not provide much insight into outcomes at the individual-level nor health and well-being-related outcomes associated with telework. Finally, Tavares [5] primarily focused on the pros and cons of telework and its proposed health effects but did not provide a broader picture of the relational components guiding the relationship between telework and worker health and well-being. Nor does the review explain the conceptual and theoretical frameworks guiding the current state of telework research today. Our review aims to address these gaps.

### Current Review

In the current review, we propose a conceptual framework for organizing and synthesizing telework and worker health and well-being research across disciplines. Our model includes predictors, mediators, moderators, and outcomes of telework at the individual worker, social and family, and organizational levels of analysis. We explain these components in-depth, followed by recommendations for future research to advance our knowledge about substantive topics and address methodological issues. We conclude the paper with recommendations for organizational policies and practices to support positive worker health and well-being. We conceptualize worker well-being broadly and consistently with a Total Worker Health^®^ definition described by Chari and colleagues [6], who defined well-being as “quality of life with respect to an individual’s health and work-related environmental, organizational, and psychosocial factors. Well-being is the experience of positive perceptions and the presence of constructive conditions and work and beyond that enables workers to thrive and achieve their full potential” (p. 590).

Additionally, in this article we focus on worker health and well-being, in lieu of work-related outcomes such as performance, due to the important associations between working arrangements and conditions and workers’ physical and psychological well-being, including chronic diseases (e.g., heart disease), pain, musculoskeletal injuries and conditions, anxiety, depression, job satisfaction, and worker engagement [7,8].

## 2. Methodology

Our review sought to investigate the various ways in which telework relates to worker health and well-being based on the definition of well-being established by Chari et al. [6] To identify articles for our review, we conducted multiple literature searches using Google Scholar and PRIMO search engines to investigate research published through January 2022. The keywords used for literature searches included: “telework”, “remote work”, “telecommute”, “telecommuting”, “occupational health”, “occupational health psychology”, “work design”, “ergonomics”, “job demands”, “job resources”, “job characteristics”, “well-being”, “stress”, “strain”, “work and family”, “health”, “physical health”, “mental health”, “sleep”, “gender”, “age”, “COVID-19” and “COVID”. Additionally, we conducted searches with Google Scholar to identify articles which cited previous telework reviews, and also reviewed articles cited within previous reviews. Finally, we contacted members of our professional networks to request published and in-preparation papers about telework. We included articles in the current review which investigated telework as defined by Allen et al. [2], even when participants’ telework was conducted after hours, on the weekend, or at a remote teleworking center. Due to language restrictions, we only reviewed articles written or translated in the English language, although we did not set restrictions regarding study region. Finally, we chose to begin our search with articles published during or after the year 2000, because prior reviews [1,2] thoroughly covered the literature prior to that time period.

## 3. Theoretical Background

There are two dominant theories in occupational health literature that facilitate our understanding of the relationship between telework and occupational health: the job-demands-resources model in occupational health psychology, and macroergonomics systems theory in ergonomics. Next, we discuss these theories and how they contribute to our model of telework and worker health and well-being.

### 3.1. Job Demands-Resources

Prior studies have relied upon the job demands-resources (JDR) model [9,10] to explain the relationship between telework and worker health and well-being [11,12]. According to this model, when individuals have insufficient resources to meet their job demands, burnout and strain result [9]. Job demands are the physical, social, or organizational components of a worker’s job that require physical or mental effort, consume energetic resources, and are associated with physiological and psychological costs such as somatic health complaints and exhaustion. Job resources are aspects of a worker’s job which fulfill basic psychological needs and may be used to alleviate job demands. Within our own model of telework and worker health and well-being, we suggest that telework is a job resource, and in particular a structural resource that may be used once or over time [13], to improve workers’ ability to meet the demands of their job. We also incorporate the notion of personal demands, the individual standards a person sets for their performance and behavior [10], and personal resources, the positive characteristics of an individual which relate to their ability to successfully impact and control their environment, which interact with the efficacy of telework as a job resource. Examples of personal demands include perfectionism and emotional instability, whereas examples of personal resources include optimism and self-efficacy.

From another perspective, investigators have focused on how the unique work arrangement created by telework presents new concerns for identifying additional context-specific (e.g., work, non-work) demands and resources [11,12]. Some evidence suggests that teleworking changes typical job demands and resources available within the virtual work context [11]. Specifically, telework has been shown to be negatively related to exhaustion, partially due to the reduction of job demands such as reduced time pressure and role conflict, and increased perceptions of job resources such as autonomy workers experience while teleworking. On the other hand, telework has been shown to be negatively related to workers’ engagement in their job, in part due to reductions in feedback and social support workers experience when teleworking, as well as increased role ambiguity, which may result when choosing to work away from one’s central organization. Given the prior discussion, we also draw upon a work design perspective of telework in which job characteristics interact with the work environment (e.g., remote work environment), and incorporate a variety of job-specific and contextual factors which may influence telework’s role as a structural resource.

### 3.2. Macroergonomics Systems Approach

Teleworkers’ health and well-being rely on a composite of job resources, such as job characteristics, workspace design, ergonomic support, and information and communication technology (ICT). Ergonomics has been defined as “the use of knowledge of human abilities and limitations to the design of systems, organizations, jobs, machines, tools, and consumer products for safe, efficient, and comfortable use” [14] (p. 4). Ergonomic science is not bound to a specific domain, but is broadly concerned with the interaction between humans and a given system, such as a given organization or the organization of one’s work [15].

Macroergonomics is the study of work systems represented through workers working together, using technology, within an organizational system [16]. This organizational system is represented through an internal environment, both physical and cultural. The effectiveness of the organization system is shaped by the design of both the technological and personnel sub-systems, and how well these components are designed in respect to one another. Within the teleworking context, macroergonomics is crucial for determining how best to implement and support ICT for work [17] and understanding risk factors for employee health and safety [18]. The interaction between the organization, employee, and the effectiveness and availability of technology is a primary contributor to successful teleworking as will be reflected throughout our review of the teleworking literature.

## 4. A Conceptual Model of Telework and Worker Health and Well-Being

Figure 1 lists the antecedents, outcomes, mediators, and moderators of telework at the individual, social, and organizational level to provide a holistic picture of factors related to telework and worker health and well-being. We developed the model below to organize the various factors associated with teleworker health and well-being based on our review of the empirical literature that we summarize in this article. We encourage other researchers to draw upon, as well as further develop, our conceptual model when pursuing future research oriented at an understanding of the health and well-being outcomes associated with teleworking.

## 5. Antecedents

In the following sections, we describe various antecedents within the telework context associated with teleworker physiological, psychological, and professional health.

### 5.1. Demographics

#### 5.1.1. Gender

There are a variety of teleworking outcomes that differ by gender. For instance, women are more likely to be expected to combine multiple roles when teleworking within the household and experience greater role conflict as they manage multiple roles at once, such as employee, partner, caregiver, or parent [19]. Additionally, women teleworking in the EU during the COVID-19 pandemic demonstrated greater odds of experiencing musculoskeletal pain and discomfort overall, were more likely to report high severity pain and discomfort and experienced more family-to-work conflict than men [20]. Furthermore, women, regardless of the presence of children in the home, reported significantly higher levels of pain and discomfort when teleworking than men with children, and those with children also reported significantly higher levels of stress than men with children. However, women without children present when teleworking reported less work-to-family and family-to-work conflict than both men and women with children. Thus, it is possible the adverse effects of teleworking for women’s physical and psychosocial health as compared to men may be due, in part, to their likelihood to assume multiple roles when teleworking [19], and these effects are exacerbated when taking childcare responsibilities into consideration. Future work should aim to separate effects on teleworker health and well-being by as a result of one’s gender, or traditional gender biases, versus as a result of one’s family structure or caregiving considerations.

Furthermore, there are other important psychophysiological differences by gender relevant to telework. For example, in a mixed-method study, men had significantly higher levels of epinephrine, more commonly known as adrenaline, in the evenings when teleworking during the day than women [21]. The authors speculate this difference may be due to men being more likely to continue to work longer into the evenings than women when teleworking, as evening levels for men were higher on days working from home than when working from the office. Qualitative findings from this study also supported this assumption but did not allow for a statistical comparison.

#### 5.1.2. Age

As the population ages and the traditional retirement age increases, employers are faced with many unique worker retention and recruiting challenges [22]. One way to effectively retain, recruit, and support older workers may be the use of flexible work practices, including telework [23]. Prior research has shown that telework usage, specifically the amount of time spent teleworking and type of ICT usage, does not vary between young and older workers [24,25]. However, older workers have reported lower ratings of self-evaluated computer skills within the telework context and willingness to telework overall [25]. Still there is little, if any, empirical investigations that have evaluated whether these factors contribute to deviations in the health and well-being of older workers.

Considering the JDR model, it is likely that older workers have different job and personal demands compared to younger workers, which may impact their well-being. For instance, older workers have a higher risk of developing chronic health conditions. Middle-age and older workers may have more eldercare responsibilities than younger workers [26]. Under these circumstances, there may be differential health and well-being outcomes between older and younger teleworkers. Future research and practice would benefit from additional studies identifying potential age differences.

#### 5.1.3. Location and Physical Environment

Research regarding differences in telework locations is relatively nascent; however, where an employee chooses to telework may influence the outcomes of that telework on the employee. For instance, home-based teleworkers experience more work-life balance support than client-based workers and those working from remote tele-centers [27]. This may be due, in part, to increased autonomy, flexibility, and a decreased commute time experienced when working from home; whereas, working from a remote tele-center may only help to reduce travel and not provide the flexibility or autonomy needed to promote work-life balance. Remote home-based workers also report higher ratings of job satisfaction than client-based workers, further speaking to the benefits of working from one’s home, specifically, versus remote work in and of itself.

Beyond the geographic location of where one works, teleworkers might also experience adverse effects as a result of the physical environment or “microclimate” of where they choose to perform their work [28]. In their short review, Bruomprisco and colleagues discussed how factors relating to the physical environment of one’s workspace, such as air quality or air circulation systems that promote proper atmospheric conditions, are associated with worker health. For instance, home-based teleworkers who lack proper air quality or humidity within their homes may experience adverse symptoms such as eye and respiratory irritation, headaches, and fatigue among other symptoms.

#### 5.1.4. Occupation and Industry

Jobs vary in the extent to which they can be performed remotely based on the nature of the tasks, work activities, and necessary setting or equipment needed for the job. Examples of jobs that cannot be done remotely/from home are those that involve handling or moving heavy objects, controlling machines or heavy equipment, operating vehicles or mechanized devices, or inspecting, repairing and/or maintaining equipment, structures, or materials [29,30]. During the COVID-19 pandemic, many workers who had not previously teleworked shifted to telework as a measure of social distancing. Using data on work activities and work context in the Occupational Information Network, research during COVID-19 classified 37% of jobs as work that can be done at home [29].

Occupation and industry are associated with whether individuals engage in telework and/or the extent of telework. For example, research has also shown that some industries are more supportive of telework, either because they have the infrastructure available to support telework (e.g., IT) or a large proportion of workers in occupations that are conducive to telework (e.g., professional and technical services, educational services, and finance and insurance [31]). There may be certain jobs that can be done remotely, but the organizational or industry norm does not generally support telework. Industries that saw the greatest increase in telework during the COVID-19 pandemic included educational services, finance and insurance, management of companies and enterprises, IT, scheduled air transportation, and professional and technical services. In October 2020, the Pew Research Center [30] conducted a survey about telework among workers in nine industries and found that the majority of workers in four of those industries indicated that their job can be done from home: 84% in banking, finance, accounting and real estate, 84% in information and technology, 59% in education and 59% in professional, scientific and technical services. However, there has not been any research to date comparing worker well-being across occupations and industries in relation to telework. The process by which occupation relates to worker health and well-being is explained in the next section.

### 5.2. Job Characteristics

Job characteristics can vary on many dimensions. Psychological research has identified some job characteristics that affect worker psychological processes, including motivation, experienced meaningfulness at work, and job satisfaction [32]. Job characteristics, such as autonomy, participation in decision-making, and social support, are shown to predict teleworker work-related well-being [33]. Vander Elst et al. originally investigated whether these components mediated the relationship between the extent of telework and four indicators of work-related well-being (emotional exhaustion, cynicism, work engagement, cognitive stress complaints). No direct or indirect relationship was found between the extent of telework and work-related well-being (with the exception of social support, discussed later), as all of the included job characteristics were directly and beneficially associated with work-related well-being. The authors suggested these results reflect that how telework impacts employee well-being depends on how the work itself is organized as well as the organizational practices in place meant to support teleworking arrangements.

The role of job characteristics in relation to worker health and well-being has also been studied via qualitative accounts among high-intensity teleworkers in China during the COVID-19 pandemic [34]. In Wang et al.’s mixed method study, workers most frequently referred to the role of job characteristics, such as increased job autonomy and perceptions of work overload, in relation to their work productivity and ability to achieve work-life balance when teleworking. Participants also mentioned the adverse characteristics of teleworking, which they often avoided when working from their central organization, such as monitoring from their supervisors, increased meeting frequency, and increases in loneliness and the need for social support.

Furthermore, participant survey responses from Wang and colleagues’ [34] study reflect social support as associated with lower levels of procrastination, ineffective communication, work–home interference, and both social support and job autonomy are shown to be associated with lower levels of loneliness. Conversely, participant workload and the extent to which they perceived monitoring from their employers is linked to higher levels of work–home interference. These results are similar to those in Pulido-Martos et al. [35] in which survey responses from workers teleworking at various intensities demonstrated a positive relationship between the levels of social report workers received and their levels of vigor, considered a personal resource while working. Thus, as the job-demands resource model may assume, social support and job autonomy are job resources which alleviate challenges associated with remote working, and workload and monitoring are seen as job demands which compromise employee well-being. However, the authors also found that workers experienced lower levels of social support when teleworking full-time, versus hybrid or in-office workers, and subsequently lower levels of vigor than workers who had a hybrid or face-to-face work arrangement. Nonetheless, these findings support a work design perspective of telework in which job characteristics are antecedents to both employee performance and well-being [34]. Across studies, high levels of social support and autonomy help in overcoming potential challenges of teleworking, such as feelings of loneliness, and in turn are associated with teleworker performance and well-being.

### 5.3. Extent of Telework

In the early stages of telework research, studies often focused on the differences amongst teleworking and non-teleworking workers [2]. Only in recent decades have investigators begun addressing the extent of telework, or the average amount of time an individual spends teleworking as a proportion of their working week [36]. The extent of telework has become a common denominator across teleworking studies. The underlying notion of these investigations is that a worker who teleworks once a week is likely to have differential experiences than a worker who spent their full week teleworking [2]. Along these lines, the extent of telework has been shown to be a significant predictor, as well as moderator, of multiple worker health and well-being outcomes in addition to work-related outcomes such as job performance.

A number of studies have shown a positive association between the extent of telework and work-related well-being. For instance, telework is positively related to job satisfaction [36,37,38,39], and especially for those who telework a moderate amount (approximately 40% of one’s total working hours) versus more or less frequently [36]. This curvilinear relationship between the extent of telework and job satisfaction has been replicated within multiple studies [36,37]. However, it is important to note that job satisfaction does not decrease dramatically for those who telework more than a moderate amount, but only tapers slightly or plateaus at higher teleworking intensities. One, among many, explanations for the positive association between the extent of telework and job satisfaction is decreased interruptions within one’s home and limited exposure to organizational politics [40].

Furthermore, work by Golden [37] indicated that the utilization of telework is related to higher quality relationships with leaders, lower quality relationships with coworkers (though coworker relationship quality was not associated with work-related well-being), and decreased work–family conflict. These relations, in turn, were positively related to job satisfaction and, notably, grew in strength with the amount of time spent teleworking. Conversely, limited face-to-face interactions and social isolation may contribute to the plateau in job satisfaction at higher intensities of teleworking [36].

The extent of telework also has consequences for employee health and well-being. In a longitudinal study using employee health claims, health risk assessment (HRA) data, and employee remote activity hours, showed the number of teleworking hours has implications for employee health [41]. Employees who worked remotely eight hours or less a month were more likely to reduce their risk of depression over time than non-telecommuters. Furthermore, teleworking more hours per month was associated with lower instances of alcohol abuse and tobacco abuse as well as lower health risks overall as calculated by participant Edington risk score. The opposite was found for stress: the more hours employees telecommuted, the greater the risk for overall perceived stress. However, in a different study, Vander Elst et al. [33] did find that the extent of telework was indirectly related to employee emotional exhaustion, cynicism, and cognitive stress complaints when mediated by social support.

Finally, one consideration when evaluating the extent of telework for worker health and well-being is the degree to which our understanding is constrained by the modest time spent teleworking by past employee samples; wherein, empirical work including participants who telework at high intensities (e.g., more than 2–3 days a week) is less common. However, the COVID-19 pandemic has provided researchers with the opportunity to evaluate worker health and well-being as a result of high-intensity telework, with many workers transitioning to full- or close to full-time telework, especially during the early stages of the COVID-19 pandemic. We expect upcoming work to provide further clarification on the role of high intensity telework for worker health and well-being.

### 5.4. Individual Differences

#### 5.4.1. Personality Characteristics

Despite differences in work performance across workers with various personality characteristics [42], little research has addressed the role of personality in predicting teleworking workers’ health and well-being outcomes. However, extant literature does tell us that personality plays a role in determining teleworker health [43]. In two field studies, workers who were high in emotional stability, and also reported high autonomy, experienced the lowest levels of psychological strain. Overall, there was a negative relationship between the extent of telework and strain for these workers, which the authors attributed to these workers being able to best meet their needs for autonomy, relatedness, and competence through remote work, which in turn reduced perceived strain. The opposite associations were seen for workers low in emotional stability. In spite of reporting high levels of autonomy, these workers were more susceptible to strain, and were likely to experience more strain as the number of teleworking hours increased.

The need for autonomy is also related to higher levels of job satisfaction among teleworkers versus non-teleworkers [42]. In their study, O’Neill et al. evaluated personality characteristics as predictors of teleworker versus non-teleworker performance and job satisfaction (i.e., organization, diligence, sociability, need for autonomy, and need for achievement). Although only the relationship between the need for autonomy and job satisfaction was significantly higher for teleworking employees, there was some evidence that sociability has a negative association with teleworker job-satisfaction, though this relationship was not statistically significant.

Finally, recent research has investigated proactive personality, which refers to one’s “tendency to identify opportunities for change, and to act on them until they bring about the desired change” [44] (p. 9). A recent study by Abdel Hadi et al. demonstrated how having a proactive personality is a beneficial antecedent to teleworker health and well-being. In their daily diary study, Abel Hadi and colleagues surveyed German employees about facets of their personality and occupational characteristics as well as their daily perceptions of job and home demands and the extent to which they engaged in leisure crafting during a mandated lockdown during the COVID-19 pandemic in which workers were required to remain in their homes. Petrou and Bakker [45] referred to leisure crafting as a mechanism for proactively pursuing leisure activities aimed at reaching a goal, human connection, learning, or personal development. In Abel Hadi et al.’s study [44], across participants, those who scored higher on a measure of proactive personality also reported more engagement in leisure crafting and less job and home-demands, which in turn were associated with lower levels of emotional exhaustion and better job performance. Thus, as a whole, we might consider both one’s emotional stability and proactivity personality as resources which limit the adverse effects of job demands, such as emotional exhaustion, within the telework context.

#### 5.4.2. Boundary Preferences

Prior research has established that telework (particularly when working at home) blurs boundaries between work and home [2,46]. Boundary management refers to the process by and extent to which individuals separate their home (non-work) responsibilities from their work responsibilities or vice-versa.

Within the telework setting, employee boundary strategies range from those which are highly segmented, such as having a separate office for remote work or setting strict working hours within the home, to strategies that integrate or combine roles (e.g., take breaks from work to assist with childcare or perform household chores or work in a shared room in order to increase time spent with family or others (e.g., roommates, partners) within the home [47]. Thus, workers may use boundary management strategies to categorize role demands into the domains of either their work or home [48].

Based on the JDR model, one’s ability to effectively manage their boundaries between work and home, when also working at home, may be seen as a personal resource which relates to workers’ health and well-being. A worker’s preferred boundary management strategy is related to various occupational health outcomes for workers. For instance, workers with integration-based strategies tend to report more family-to-work conflict in which family roles and responsibilities interfere with the work domain [48]. Similarly, Allen et al. [49] found that segmentation preferences were positively related to work/non-work balance, and the same association remained consistent over three months. Additionally, those with greater boundary permeability, and especially when nonwork behaviors are interrupted by work-related responsibilities such as by working after hours or on the weekends, are more likely to have increased work-to-family conflict within the telework context [47,50].

Considering the empirical literature, workers who implement segmentation boundary strategies are at a lower risk of adverse occupational health outcomes. In a qualitative study of 40 teleworking workers, Basile and Beauregard [51] identified physical, time-based, behavioral, and communicative strategies that successful teleworkers implemented within their home. These strategies included having a separate office or space to be used for work, engaging in activities which signaled the end of the working day, switching off email or work phones after work hours, and informing friends and family of their boundary expectations regarding interruption during the work week.

### 5.5. Economic Factors

#### 5.5.1. Commute Time

Reduced commute time has not been shown to be a motivator for teleworking [1] although reductions in driving time, in particular, for teleworkers is linked to reductions in commute stress [26]. However, it is possible that reductions in commute time for teleworkers may benefit workers’ physical health. In general, passive commute distance, or distance commuted by vehicle, has a negative relationship with several physical health indicators such as physical activity and cardiorespiratory fitness, and is adversely associated with one’s BMI, waist circumference, blood pressure, and metabolic risk [52].

However, the benefits associated with commute time may only arise for workers teleworking a full workday from home. For teleworkers choosing to conduct their work from libraries, cafes, or similar locations, there is a likelihood to avoid peak-hour travel; however, it is unlikely that workers will experience significant reductions in travel time [53]. Furthermore, workers who attend their central workplace before teleworking the remainder of their workday are also unlikely to experience reductions in travel time or avoid peak-hour travel, which may also lend to differences in perceived stress between full- versus part-day teleworkers [26]. Full-day teleworkers are also more likely to rely on active modes of transportation when leaving their homes (e.g., walking or biking) [54], which might serve to benefit employee cardiovascular health.

With regard to work-related outcomes, reductions in commute time have also been linked to increased productivity. However, this increase in performance is speculated to be due to longer working hours, as teleworkers may continue working into time typically reserved for driving or other forms of travel [1,2,55]. Notably, longer working hours are, in turn, adversely associated with coronary heart disease and depression [56]. Thus, future research should identify whether reductions in commute time as a result of teleworking, and in particular reductions in passive commute time, has an effect on worker health and health behaviors, and under what conditions we may expect reductions in commute time to relate to positive health outcomes. For instance, we might expect a positive relationship between a reduction in passive commute and cardiorespiratory fitness if workers choose to use the time previously allocated for a passive commute for beneficial health behaviors such as physical activity or for preparing healthy meals.

#### 5.5.2. Economic Resources

In terms of cost savings and economic resources related to telework, much of the empirical literature is rooted within the overall business impact. Some research has addressed the role of the economic resources for the employee within the teleworking context. For instance, prior research has shown that although the reduced commute time and availability of ICT play a minor role in predicting the choice to telework [57], reductions in commute time might also reduce or alleviate the financial strain of paying for gas, road tolls, and public transit. Although some research counters this assumption by showing that teleworkers drive more non-work miles than non-teleworkers [58,59,60]; often replacing commute miles with nonwork trips such as to the grocery store or to run other errands that would often take place alongside one’s daily work commute. When considering an occupational health perspective, future research might consider the impact that one’s financial and personal resources have on the effectiveness of telework utilization. Factors such as the availability of updated computer technology, a high-speed wireless connection, who bears the cost of technology (i.e., the employer or employee), and whether workers’ have sufficient space within the home for a separate workspace might contribute to workers’ well-being.

### 5.6. Ergonomic Resources

#### 5.6.1. Training

Organizational concern for worker health and safety should not differ between the home or traditional office space [61]. However, there is little empirical literature related to telework and ergonomic factors. Teleworking employees often have little awareness and knowledge related to ergonomic and safety issues within their homes [17]. In addition, many companies lack sufficient regulation and policies regarding the set up and ergonomic evaluation of in-home and remote workspaces [62]. Compounding these issues is the lack of reliable injury frequency and severity reporting for teleworking employees [17,26]. These factors are surprising, as in-home and remote workers are still performing work-related duties and injuries incurred while working may still be covered through Worker’s Compensation [62].

When teleworkers are not provided with the proper ergonomic training and resources, such as an organization-provided ergonomic workstation, sufficient technical assistance to evaluate and adjust one’s workstation as needed, and training over best ergonomic and/or telework practices, they incur increased musculoskeletal and psychosocial risks [62]. Prior research has shown that teleworkers often set up their own telework spaces and engage in risky behaviors such as working from the couch or other uncomfortable workspaces. Harrington and Walker [63] state that without appropriate training, workers are likely to be unaware of the risk that these and other home-working behaviors may have on their potential to develop chronic musculoskeletal disorders. Accordingly, home office ergonomics training has been shown to improve workers’ knowledge, attitudes, and ergonomic practices. Furthermore, ergonomics training is associated with less pain and discomfort for workers receiving the training. In addition, workers who receive telework-specific training adjust faster to teleworking than those who do not receive training [55]. This indicates that ergonomics training may have beneficial effects for both teleworker physical and psychological health.

#### 5.6.2. Information and Communication Technology

Based on the JDR model, ergonomic training and sufficient computer technology and assistance are job resources which can alleviate adverse occupational health outcomes. Suh and Lee [64] show the interaction between one’s technology as well as how the characteristics of their job can lead to technostress, or stress caused by information computer technology. Technostress can then lead to reductions in job satisfaction. For example, in Wang et al.’s [34] study of teleworkers in Japan during the COVID-19 pandemic, the top inconveniences reported were related to insufficient technology and desktop space as well as slow internet speeds.

Furthermore, as the prevalence of teleworking increases, so might the prevalence of virtual work meetings among members of an organization. With this in mind, researchers have begun to note the association between forms of ICT usage for conducting meetings and tasks among remote workers and the potential effects for employee health and well-being. For example, virtual and video-based meetings (e.g., Zoom) can lead to fatigue [65], and subsequently reduce their engagement and likelihood to voice concerns in the workplace [66]. These effects are particularly pronounced for women and newer workers who may be more concerned with impression management. When evaluating these results through the lens of the macroergonomics systems approach, we can further understand how the organization of technology and personnel subsystems can affect the trajectory of worker health and well-being outcomes.

### 5.7. Organizational Factors

#### 5.7.1. Support for Telework

The amount of support an individual receives from their organization also plays a role in facilitating employee well-being. Using a socio-technical systems approach, similar to the macroergonomics systems approach, Bentley et al. [67] hypothesized that when employees perceive support by the organization addressing technical, person, and organizational subsystems, it is likely that teleworking will relate to better worker well-being (i.e., psychological strain, social isolation, and job satisfaction). The authors included both measures of organizational social support and telework support. Telework support is organizational practices which support the effective practice of teleworking. These practices include trust and resources provided by one’s supervisor for teleworking, as well as the amount of technical support provided to the teleworker. In line with their hypothesis, both organizational and telework support significantly increased job satisfaction while also decreasing psychological strain for teleworking employees. In addition, organizational support was associated with lower levels of social isolation. These results support the fundamental assumptions of the socio-technical systems and macroergonomics systems approach which suggest that the effectiveness of telework is associated with how well the organizational, person, and technical systems of the organization are aligned.

#### 5.7.2. Formality

The formality of one’s telework arrangement established by organizational policies or arrangements with one’s supervisor may contribute to a worker’s overall telework experience. More flexibility and informal arrangements may increase worker’s sense of autonomy and perceptions of support versus formal policies which delegate when, where, and how one engages in telework. Accordingly, Kossek et al. [47] recommended future researchers to identify the formality of workers’ telework arrangement in their studies. In their study, having a formal telework policy was related to higher performance ratings. However, having a formal policy was also associated with higher levels of depression, with the exception of women with children. The use of formal telework arrangements have also been associated with higher reports of employee job satisfaction when compared with informal arrangements [68] although these differences were only statistically significant for women. Following Kossek et al.’s [47] recommendation, future work should continue to balance workers’ perceptions of flexibility against the formality of their telework arrangements.

### 5.8. Summary

Our review indicates there are a variety of factors which relate to whether and why individuals may engage in telework. For example, gender is associated to some degree with teleworkers’ comfort and psychological well-being while teleworking. Men typically have healthier teleworking experiences than women, although these findings seem to be related to work and family roles rather than gender per se. For example, the results indicated that women are more likely to juggle multiple roles (e.g., worker, partner, parent, etc.) when teleworking [19], which is consistent with traditional gender role norms [69]. Other factors, such as reductions in commute time, location of where the work is performed, occupation and industry, job characteristics organizational support, and access to working computer technology and ergonomically designed workstations also have an impact on workers’ telework experiences. Outcomes associated with telework are described in the next section.

## 6. Outcomes

### 6.1. Health Outcomes

#### 6.1.1. Physical Health

Research evaluating the physical health of workers as a result of telework is still sparse and equivocal. According to Henke et al. [41], the extent of telework is beneficially associated with employee health, with teleworking employees having a lower overall risk of poor health than non-teleworkers. Similarly, in a study by Lundberg et al. [21], both men and women had lower systolic blood pressure, a known stress indicator, when teleworking versus working from the main office, although this association was only significant for women. However, reduced blood pressure for teleworkers may be due to reduced physical activity, rather than reduced stress. Though limited, these results suggest there may be pros and cons for employee physical health when teleworking, and future research should aim to identify which health behaviors and job resources may alleviate demands associated with worker health and well-being while teleworking.

#### 6.1.2. Health Behaviors

Research with a specific focus on employee health behaviors within the teleworking context has only recently emerged, and only a few studies have identified the role of telework in predicting employee health behaviors such as physical activity, nutrition, and substance use. As briefly mentioned in prior sections, Henke et al. [41] found teleworking employees to be at a significantly lower risk of poor nutrition, physical inactivity, and tobacco use than non-teleworkers. Furthermore, workers performing 50% or more of their teleworking hours during traditional hours were at a significantly lower risk of alcohol abuse. On the other hand, those who telework during non-traditional hours or over the weekend were at a higher risk of alcohol abuse than both those who telework during traditional hours and non-teleworkers. More recent research conducted during the COVID-19 pandemic [70] also identified changes in substance use behavior, although it is difficult to assess the extent to which the increase in substance use was related to work (e.g., telework) or other factors associated with well-being.

In general, the positive association between telework and healthy behavior is in line with the empirical literature on workplace flexibility (as in telework), where employees reporting higher levels of flexibility also reported higher frequencies of physical activity [71]. Similarly, results from Allen et al. [72] indicated that greater flexplace flexibility (i.e., telework) is associated with less fast-food consumption. With regard to health care utilization, results from Butler et al. [73] showed insignificant differences in health care utilization between those with higher and lower workplace flexibility. Nonetheless, we did not find any articles referencing this relationship within a telework-specific context.

In relation to worker sleep health and hygiene, employees reporting higher levels of flexibility at work also reported a higher number of hours slept on average [71]. Furthermore, workers transitioning from the office to telework in Japan during a mandated COVID-19 lockdown reported getting more sleep after their transition to home-based telework [74]. Nonetheless, it is important to note that although there are proper sleep hygiene behaviors that may help workers’ well-being, and sleep duration and quality can also be affected by factors such as chronic health conditions, work schedules, presence of children in the home, and other individual differences [75]. When looking at differences in sleep duration and quality, future telework researchers should consider these differences.

#### 6.1.3. Musculoskeletal and Pain Symptoms

Much of what we know regarding musculoskeletal symptoms and telework is related to extended computer usage. Teleworkers typically rely on computer technology as their main mode of task completion, and computer use is associated with extended static postures, repetitive movements, and wrist and forearm fatigue [55]. Subsequently, these factors are associated with the development of musculoskeletal symptoms and disorders within the neck, wrist, shoulders, hands, and lower back. Although adequate computer workstations and ergonomics training may alleviate these risk factors [63], in Montreuil and Lippel’s empirical study [55], 54.5% of teleworkers experienced pain symptoms in their upper limbs, back, and neck which they contributed to inadequate computer and workstation furnishings. Furthermore, the lack of interruptions and face-to-face interaction when teleworking may lead to a reduction in work breaks or longer working hours for some workers, which may also strengthen the likelihood of developing musculoskeletal symptoms. Lastly, there is speculation that psychosocial aspects of telework including time constraints and a lack of social support may lead to the development of musculoskeletal symptoms among teleworking employees [55,76].

#### 6.1.4. Mental Health

There is an obvious lack of empirical investigations into telework that include measures of anxiety, depression, and other indicators of mental health. The research that is available is fairly equivocal. For example, Henke et al. [41] reported employees teleworking eight hours or less a week were significantly less likely to experience depression than non-teleworkers. On the other hand, Mann and Holdsworth [19] found teleworking employees to experience significantly more mental health symptoms related to stress as measured by the Occupational Stress Indicator (OSI) [77]. The difference in mental health outcomes between these studies may be due, in part, to differences in the extent of telework practiced by participants. Henke et al. [41] surveyed participants across a spectrum of weekly telework hours, whereas the participants within Mann & Holdsworth were either full-time teleworkers or full-time office workers [19].

#### 6.1.5. Psychological Well-Being

Psychological well-being refers to attitudes and experiences workers have related to their overall well-being, such as job satisfaction, life satisfaction, and burnout. Literature relating to the psychological well-being of teleworkers is also largely indeterminate, although there is a general agreement that job characteristics play a large mediating and moderating role in predicting teleworker psychological well-being. However, there is also an abundance of measures which investigators used to evaluate teleworker well-being (e.g., net-affect, psychological strain, emotional exhaustion, and job engagement), and it is this inconsistency which may be contributing to the equivocation of results regarding the relationship between telework and worker well-being. However, certain trends do emerge within the literature.

For example, Song and Gao [78] found telework to be associated with lower levels of tiredness and Sardeshmukh et al. [11] report a significant, beneficial association between telework and exhaustion, partially mediated by job demands (time pressure, role ambiguity, role conflict) and job resources (job autonomy, feedback, and job support). However, Sardeshmukh et al. [11] also reported a significant negative association between telework and job engagement, partially mediated through the same demands and resources. Similarly, Vander Elst et al. [33] did not find a direct relationship between the extent of telework and work-related well-being indicators (i.e., exhaustion, job engagement, and cognitive stress complaints), but the authors did find an indirect, negative relationship between the extent of telework and work-related well-being via lower levels of social support. Thus, teleworkers who teleworked more days a week experienced less social support, and in turn experienced higher levels of adverse well-being indicators. These results support those in Sardeshmukh et al. [11] where social support was a prominent mediator between telework and well-being, and also reflect findings that suggest that teleworkers may experience higher levels of exhaustion when teleworking at high intensities [79]. However, participants in the latter study only experienced increased exhaustion when also experiencing high-levels of work–family conflict.

In a different study, Duxbury and Halinski [80] found the extent of telework to negatively moderate the positive association between the number of total hours worked and work strain (i.e., work role overload). Telework was shown to help workers with high job demands alleviate the negative influence of those demands on their work-related well-being. This process may be due to an increase in job control when utilizing their ability to telework. These results complement reports from Perry et al. [43] in which employees high in emotional stability and high in perceived autonomy experience the least psychological strain when teleworking, regardless of the amount of time spent teleworking each week.

Finally, in a recent quasi-experimental, daily-design study, participants from Belgium who teleworked up to two days a week as part of a two-week intervention reported lower levels of perceived stress post-intervention, as well as lower levels of perceived daily stress on days when those participants teleworked [81]. Teleworking participants also reported higher levels of work engagement on days spent teleworking; however, participant overall work engagement did not change compared to pre-intervention. Considering the previous discussion, more research is needed to uncover the intricacies in the relationship between telework and worker psychological well-being. However, the majority of evidence seems to suggest the beneficial association between telework and psychological well-being may largely rely on the design of one’s teleworking arrangement, as well as the job resources in place for mitigating personal and family demands.

### 6.2. Social and Family Outcomes

#### 6.2.1. Work/Family Conflict and Balance

Research into the extent to which telework is beneficial or detrimental to balancing work and family is largely equivocal in the results. From one perspective, providing flexible work arrangements such as telework is seen as a way to increase work–life balance and reduce work–non-work conflict [2]. In particular, telework might lead to reductions in commute time, perceived flexibility over both one’s workplace and work schedule, as well as opportunities to manage familial and personal responsibilities such as a child home sick from school without having extreme disruptions from work which may help promote employees’ perceptions of work–life balance. On the other hand, telework inherently blurs the spatial boundaries between work and home, thus increasing the likelihood of work–family conflict [19]. For a review of work/nonwork outcomes and empirical literature prior to 2015, See Allen et al. [2].

Since the start of the COVID-19 pandemic, more studies have investigated work-family conflict as an outcome of telework. First, traditional gender dynamics have seemingly held during the COVID-19 pandemic with important implications for worker health while teleworking [82]. Shockley et al. surveyed heterosexual, dual-earning couples upon the beginning of the COVID-19 pandemic, as well as during a 2-month follow-up. A substantive proportion (36.6%) of couples reported maintaining historical gender norms relating to how they managed work and family during the COVID-19 pandemic. For these couples, in which the wife worked remotely and was also solely responsible for childcare without alterations to their husband’s work schedule or location, women reported significantly poorer outcomes, including the lowest ratings of family cohesion, the highest ratings of relationship tension, and the poorest reports of job performance, even when compared to women who were the sole remote worker but received at least partial assistance in maintaining childcare responsibility from their husbands, as well as those outsourcing childcare. Men in these relationships also experienced adverse outcomes and reported the lowest levels of family cohesion and the highest levels of relationship tension. Conversely, when couples chose to alternate in-person working days as well as childcare responsibilities, they also experienced the best relational and performance outcomes. Overall, results from Shockley et al. perhaps demonstrate the nuanced role of telework as a flexible work arrangement and also speak to the need for family-supportive practices such as provisional resources for childcare or flexible work schedules, which might better support women and men looking to balance both work and family.

Even so, teleworking in light of the COVID-19 pandemic may have silver linings for closing the gender gap in familial childcare overall, despite potential effects for familial strain or job performance. During the COVID-19 pandemic, teleworking fathers increased the amount of time spent engaging in childcare overall (approximately 67 min more on days spent teleworking), more closely representing the time typically spent by teleworking women [83]. Similarly, in Pineault et al.’s [84] study of dual-earning heterosexual couples, women undertook significantly more physical and cognitive household labor when both members of the couple worked outside of the home, versus when both members were teleworking. However, women still assumed 60% or more of both cognitive and physical household labor than men, regardless of where each member chose to work.

#### 6.2.2. Interpersonal Relationships

Although a primary assumption of teleworking is that it affords employees more autonomy and flexibility, authors have discussed a paradox in which the increased control teleworking affords is undermined by a negative association with outcomes in the social domain. For example, Mann et al. [19] present a social comparison effect in which participants report a tendency to look to others in order to derive behavioral norms [85]. The reduction in face-to-face communication and a reliance on ICT can, therein, lead to adverse social effects for teleworkers. Subsequently, teleworking employees may experience more negative emotions than office-workers, largely due to feelings of loneliness and social isolation.

However, in other cases, a lack of face-to-face interaction may be seen as a benefit to telework [40,86]. In Collins et al.’s [86] qualitative study, participants welcomed the opportunity to be removed from the social environment of the office. For these employees, telework afforded them the opportunity to avoid social conflict or negative office relationships and, in turn, foster positive work relationships. However, the longer and more frequent employees teleworked, the less connected they felt with their office-based co-workers and oftentimes did not forge new office-based relationships beyond those established prior to the commencement of their telework arrangement. Thus, over time, teleworking employees may begin to experience a reduction in their social support network. Considering this notion, both researchers and practitioners should not only account for the extent of telework, but also how long one has been teleworking when evaluating interpersonal outcomes associated with telework.

### 6.3. Work-Related Outcomes

#### 6.3.1. Job Satisfaction

Research investigating the association between telework usage and job satisfaction is vast. There seems to be consistent agreement that telework is associated with increased job satisfaction [39,87,88]. In a meta-analysis of 28 studies, Gajendran and Harrison [88] indicated a positive relationship between telework usage and job satisfaction. However, Golden and Veiga [36] reported a curvilinear association between the amount of time spent teleworking and job satisfaction, such that the positive association between these variables plateau at higher levels of telework hours.

Research revealed that these relationships are defined by a variety of mediators and moderators. For instance, Gajendran and Harrison [88] also showed that perceived autonomy fully mediates the relationship between telework and job satisfaction, while both work–family conflict and relationships with supervisors are partial-mediators. In addition, the curvilinear link between the extent of telework and job satisfaction in Golden and Veiga’s [36] study was moderated by both task interdependence and job control. Additionally, Golden [37] showed leader–member exchange quality, team-member exchange quality, and work–family conflict to mediate the curvilinear relationship between the extent of telework and job satisfaction.

The importance of job characteristics in helping to define the relationship between telework and job satisfaction has been modeled in numerous other studies. For instance, participants who were teleworking full time from India during the COVID-19 pandemic reported the highest levels of job satisfaction when also reporting high levels of job autonomy and family supportive supervisory behaviors (FSSBs) [89]. Specifically, when participants reported high levels of job resources, they also reported high levels of work–life-balance, subsequently leading to higher levels of job satisfaction. Furthermore, participants reported the highest levels of job satisfaction when they not only perceived high levels of job resources, but also reported having at least some experience teleworking prior to the COVID-19 pandemic.

Finally, Fonner and Rollof [40] show reduced disruptions from colleagues and office politics when teleworking to positively impact teleworker job satisfaction. In addition, media richness [90] and stress related to technology usage (i.e., technostress) [64] also positively contributes to teleworking employees’ job satisfaction. These latter components play a part in our understanding of the telework and job satisfaction relationship through the macroergonomic perspective. Thus, organizations may not expect positive work and health outcomes without taking both the technological and human subsystems into consideration.

#### 6.3.2. Absenteeism/Presenteeism

Although limited, research has shown that providing workers with the option to work away from the central office is associated with reduced absenteeism. However, this reduction in absence may be tied to a health trade-off for employees who continue to telework when sick (i.e., presenteeism). For instance, while an employee may choose to telework when feeling unwell in order to prevent the spread of communicable disease and also avoid absence at work, it is likely that their work performance may be less than optimal and could slow their recovery and thus may be less effective when choosing to continue working while feeling unwell. While this process, referred to as presenteeism, is often a challenge for both office and home-workers, the option to telework exacerbates the opportunity for presenteeism by removing the physical presence of the employee [19].

Steward [91] also showed that this form of invisibility was related to workers’ likelihood of working while sick. Interview participants and survey respondents indicated that their lack of presence in the office made it harder for them to justify the need to take a formal sick day, and often employees chose to continue working despite malaise. Mann and Holdsworth [19] suggest that employees may also feel lucky, or “privileged”, to work from home and choose to work through sickness in order to preserve their opportunity to telework. Thus, it is not possible to interpret reductions in absenteeism as an indication of positive health status among teleworking employees [91].

### 6.4. Summary

There is still much to be known about the health effects of telework. Much of the literature about physiological and musculoskeletal outcomes of telework is indeterminate. Telework has both beneficial and adverse effects for worker health and well-being. Telework outcomes seem to be regulated by working context and job characteristics, including autonomy and support. The extent of telework also plays a primary role in predicting the job satisfaction and overall health of teleworkers. Next, we review the moderators and mediators which help to define our understanding of what happens to employee health when one utilizes telework.

## 7. Mediators

### 7.1. Job Characteristics

Much of what we know relating to the mediating role of job characteristics in the relationship between telework and worker health and well-being is in relation to worker autonomy. Telework is directly associated with higher perceived autonomy, or control over how one completes their work [88], and perceived autonomy is among the strongest job characteristics when explaining the relationship between telework and employee outcomes. Gajendran and Harrison found autonomy to fully mediate the positive effects of telework on job satisfaction and partially mediate the impact of telework on employee stress (i.e., role stress). Considering the JDR framework, autonomy also partially mediates the impact of the extent of telework and both exhaustion and job engagement [11]. Sardeshmukh et al. suggest that the mediating role of autonomy is due to the lack of constraints linked to office routine, the ability to navigate when tasks are completed during the day, and potentially less managerial oversight. These components allow employees to conduct their work tasks in line with their own preferences, reducing exhaustion and alleviating psychological strain.

### 7.2. Social Context

#### 7.2.1. Relationship Quality

Despite previous findings regarding the isolating impact of telework on employee well-being [19], Gajendran and Harrison [88] found a positive relationship between telework and employee–supervisor relationships. In their meta-analysis, the quality of the teleworker–supervisor relationship was shown to partially mediate the relationship between telework and job satisfaction and turnover intentions. Thus, across studies, teleworking employees reported a beneficial impact of teleworking on relationships with their supervisors, and subsequently greater reports of job satisfaction and lower reports of turnover intentions.

The importance of workplace relationships for teleworking employees was also highlighted in Golden’s [37] study, wherein the quality of exchanges with one’s manager, coworkers, and family (i.e., work-family conflict) mediated the curvilinear relationship between the extent of telework and job satisfaction. Specifically, job satisfaction increased when workers reported positive relationships with their managers and team-members before plateauing or slightly decreasing among those reporting the strongest relationships. For family relationships (i.e., work–family conflict) a higher level of telework intensity was related to decreased work-family conflict, which in turn was associated with higher levels of job satisfaction, with a slight tapering when workers reported low levels of work–family conflict.

These results are reflected in other studies in which high quality superior–subordinate relationships are related to higher levels of job satisfaction, and are also moderated by the level of telework intensity [92]. While these results are largely in favor of increased telework intensity, limited face-to-face interactions and social isolation may contribute to the plateau in job satisfaction at higher intensities of teleworking [36].

#### 7.2.2. Social Support

In contrast to the findings that telework is related to a more positive employee–supervisor relationship quality, Sardeshmukh et al. [11] found that the extent of telework was related to reductions in social support, and subsequently reduced worker engagement. This relationship may be due, in part, to reduced media richness and an increased physical distance between teleworking employees and coworkers. Media richness refers to how effectively a variety of ICT transmit social cues and mitigate uncertainty and equivocality between users [93]. ICT, such as videoconferencing, are higher in media richness than standard text communications (i.e., email) as there is at least some transmission of social cues [2]. Sardeshmukh et al. [11] suggest that reducing employee perceptions of isolation and loneliness through more rich communication media may increase perceptions of social support and could mitigate the negative impact of high intensity telework on work engagement.

#### 7.2.3. Social Isolation

Building on results from their study in which organizational and telework support were associated with more positive employee well-being outcomes, Bentley et al. [67] furthered their investigation by evaluating the mediating role of social isolation between support and employee well-being. Social isolation was found to partially mediate the relationship between organizational support and employee job satisfaction and psychological strain. Thus, when organizational support is insufficient, the negative influence of social isolation associated with the use of telework can increase psychological strain and reduce job satisfaction. Organizations should provide support by means of face-to-face interaction and ensure employees have access to sufficiently rich media for interacting with other employees in order to combat the effects of social isolation when teleworking [19].

### 7.3. Summary

The role of job characteristics and the social context of work as mediators within the telework and worker health and well-being relationship is still a relatively young vein of research. While we know that both autonomy and relationship quality play a supportive role in the relationship between telework and employee well-being, reductions in perceived social support may contribute to adverse outcomes. In addition, social isolation can undermine the positive effects the utilization of telework has for employee well-being. As research continues to unpack the job characteristics and social contexts which contribute to the telework and worker health and well-being relationship, there is a likelihood that other job characteristics and social components play a mediating role when determining employee outcomes.

## 8. Moderators

### 8.1. Gender

Previously, we described gender as an antecedent. Given that women are more likely to assume multiple roles when teleworking [19], gender is likely to act as both an antecedent and moderator in the relationship between telework and worker health and well-being. For instance, during the COVID-19 pandemic, women spent more time teleworking overall, but also spent more of that time completing work in the presence of children and attending to housework than men [94]. Thus, as women participate in home-based telework, the beneficial impact of teleworking for work–life balance may be attenuated by way of reinforcing gender roles [95], with teleworking women reporting less work–life-balance than non-teleworking employees [96].

Meanwhile, men are more likely to work within their roles independently and experience less stress and negative affects while teleworking [19,69,78,83]. For instance, teleworking mothers during the beginning of the COVID-19 pandemic (April–May 2020) reported higher rates of anxiety, depression, and loneliness than teleworking fathers, who actually experienced reduced anxiety when teleworking from home [83]. Men have also shown higher ratings of mental and physical health while teleworking in general, although these differences are not statistically significant when compared with traditional office workers [19].

### 8.2. Extent of Telework

Extant literature shows that the extent of telework also acts as a moderator among a number of occupational health outcomes. For instance, the number of hours telecommuted moderates the relationship between total working hours and work strain measured through role overload [80]. Similar conclusions were found in Gajendran and Harrison’s meta-analysis [88], wherein telework intensity beneficially moderated the relationship between telework and role stress. Higher intensity teleworkers experienced reduced role stress. In addition, high intensity teleworkers also experienced reductions in work–family conflict. A number of other studies [36,92] discuss the moderating effect of the extent of telework on various work-related outcomes. For a more thorough review of these articles, we refer readers to Allen et al. [2]

### 8.3. Job Characteristics

#### 8.3.1. Autonomy

Autonomy also serves as a moderator within the telework and worker health and well-being relationship. As previously discussed, prior research reports that there is a curvilinear link between the extent of telework and job satisfaction [36,37,92]. This work has also established the role of job discretion and the amount of autonomy an employee has in how they perform their jobs as a moderator within the relationship between extent of telework and job satisfaction [36]. Individuals who have higher levels of job discretion also show a tendency to have higher levels of job satisfaction across the extent of telework spectrum.

Additionally, job autonomy moderates the relationship between telework and work–family conflict, such that individuals with higher job autonomy experience less work–family conflict in the general context [97]. Interestingly, higher levels of job autonomy do not lead to a faster decrease in work–family conflict per additional hour of telework each week. In fact, individuals with lower autonomy see a more distinct decline in work–family conflict in relation to the extent of telework. Golden et al. suppose that this differential decline in WFC is potentially due to individuals with a lower job autonomy taking advantage of the saved time due to extensive telecommuting (e.g., reduction in commute times) in order to reduce WFC.

Finally, autonomy plays a role in determining stress outcomes for teleworking employees. Perry et al. [43] showed that when individuals have high levels of emotional stability and high levels of autonomy, they experience less strain overall regardless of the extent of telework. Conversely, when individuals are low in emotional stability and high in autonomy, they are predisposed to higher levels of work stress, leading to strain, which is likely to increase as they telework more hours.

#### 8.3.2. Flexibility

Much of what we know regarding the positive benefits of flexible work arrangements (i.e., telework, remote work) on employee health and well-being is drawn from the general workplace flexibility literature [71,73]. Although autonomy seems to play a strong role in determining teleworker health outcomes, less is known about flexibility components specific to telework. What we do know is that transitioning to telework has been reported as leading to greater perceived flexibility for employees [98]. In their qualitative, quasi-experimental study with employees from IBM, Hill et al. found that the transition to telework from office work increased employees perceived flexibility, which therein increased their employee’s personal/nonwork life.

Golden and Veiga [36] included a measure of flexibility, work-schedule latitude, in their investigation of the moderating effects of work characteristics on the curvilinear relationship between extent of telecommuting and job satisfaction. Although there was no significant moderating effect found in their analyses, the authors stipulated that as their sample included salaried professionals, it may be that flexibility in scheduling their work tasks is a common aspect of professional work and is not easily reflected in measures of job satisfaction.

### 8.4. Task Characteristics

Task characteristics, specifically task interdependence, play a role in moderating the relationship between the extent of telework and job satisfaction [36]. Task interdependence refers to the extent an individual is relied upon, or relies on others, to complete their job tasks [99]. When teleworking, individuals whose work is highly interdependent may experience more frustration due to continuous back and forth communication with other members of their organization needed to complete their own work tasks [100]. Golden and Veiga [36] reported that individuals with lower levels of task interdependence typically have higher levels of job satisfaction. This relationship follows the typical curvilinear trend found between extent of telework and job satisfaction. Specifically, individuals with higher task interdependence reflected a slower increase in job satisfaction and the differential increase between job satisfaction for those with low versus high task interdependence was more pronounced at a higher extent of telework.

Task interdependence might also influence levels of exhaustion among teleworkers. During the COVID-19 pandemic, many workers experienced daily task setbacks (e.g., changes to the implementation or requirements of their work due to the pandemic). Chong et al. [101] demonstrate how day-to-day, work-related setbacks specific to the pandemic are associated with higher levels of employee exhaustion at the end of the workday, and these associations are further exacerbated when employees engage in highly interdependent work. However, when employees reporting high levels of exhaustion at the end of their workday also reported receiving organizational telework task support, they did not report significant levels of withdrawal from work on the following workday. Conversely, when employees reported low levels of organizational telework task support, there was a significant, adverse association between end-of-day exhaustion and withdrawal from work on the following workday.

To date, these are the only empirical investigations we have found that investigated the impact of task characteristics on the link between telework and worker health and future research would benefit from additional studies evaluating telework, task characteristics, and worker health and well-being. For instance, future research might utilize O*Net, an occupational information database, to evaluate common work activities and their subsequent effects, performed by workers within their samples.

### 8.5. Voluntariness

Both formal and informal telework arrangements might benefit workers when they have a choice about whether or not they have opportunities to telework. For instance, voluntary telework, versus mandated or involuntary telework (as many workers experienced amid the COVID-19 pandemic), is associated with higher levels of job satisfaction and lower levels of turnover intentions and perceived stress [102]. Furthermore, voluntary telework supports employee perceptions of autonomy by allowing them to control their desired degree of integration and segmentation between work and family and nonwork domains [103]. This is consistent with past research that underscores the importance of giving workers autonomy or control over their jobs [8].

### 8.6. Boundary Preferences

Boundary preferences (i.e., preferences for segmenting or integrating work and non-work responsibilities) might also play a moderating role in the relationship between telework and worker health and well-being. Workers’ preferences for separating work and non-work experiences (i.e., segmentation) or combining work and non-work roles (i.e., integration) can shape their experiences and outcomes associated with telework. For example, Derks et al. [104] investigated boundary management preferences as a moderator in relation to work-related smartphone use, work–family conflict, and family role performance. Their results found no association for segmenters, but integrators experienced less work–family conflict and better family role performance.

### 8.7. Summary

Job characteristics, and especially autonomy, are important for fostering positive worker health and well-being. Job autonomy is related to better job satisfaction, less work/family conflict, and reduced worker stress. There is also some evidence for a beneficial impact of perceived flexibility, although the current state of evidence is equivocal. Task characteristics, such as task interdependence, also play a role in determining the directions of employee outcomes when utilizing the option to telework, although further investigation is needed to holistically identify characteristics beyond task interdependence which contribute to worker health and well-being.

## 9. Discussion

Overall, what we know about the relationship between telework and worker health and well-being is variable and seemingly dependent on a variety of job characteristics and contextual and technological factors. Understanding the influence of job demands and resources is integral to understanding the relationship between the utilization of telework and employee health and well-being. However, the extant literature has also demonstrated the importance of designing sub-systems that complement one another in order to obtain successful telework outcomes. When organizational, person, and technological subsystems are designed thoughtfully and intentionally, we may expect not only better productivity, but also better worker health and well-being.

Within our review, we identified a number of antecedents, outcomes, mediators, and moderators at the organizational, job, work/family/life, and individual level that explain the relationship between telework and worker health and well-being. Although some individual characteristics such as gender and personality help to predict teleworker health outcomes, both job characteristics and organizational support and practices also play strong roles in predicting teleworker well-being. The extent of telework is also a primary factor related to worker outcomes, with employees teleworking approximately 40% of their working hours experiencing the most favorable outcomes. Furthermore, job characteristics such as autonomy serve as important mediators and moderators within the telework and worker health relationship. Jobs characterized as having more autonomy and control are associated with better worker outcomes, and these effects also hold true regarding whether workers choose to engage in telework. We also discuss how the social context, wherein telework is performed, helps to further define the telework and worker health and well-being relationship. Workers’ relationships with supervisors, coworkers, and family members, as well as feelings of social isolation, can either benefit or detract from their health and well-being when teleworking.

With regard to the physical health and psychological outcomes of teleworking, much of the literature is equivocal. Though limited, the available research suggests that teleworking leads to positive health outcomes such as lower blood pressure and decreased health risks in some samples. Although, working longer or nonstandard hours due to the increased control and flexibility which telework provides may undermine these outcomes by elevating stress levels. In addition, exposure to extended computer usage and poorly designed workstations can lead to musculoskeletal and pain symptoms in teleworking employees. Preliminary evidence provides support for the utilization of telework in increasing positive employee health behaviors such as physical activity, sleep, and nutritional choices.

However, our understanding of the mental health outcomes related to telework is less clear. One challenge associated with trying to understand outcomes related to telework during the COVID-19 pandemic is that it is difficult to identify specific outcomes associated with telework and other factors that have co-occurred during this time (e.g., remote school for school-age children, workers with partners both teleworking in the same space, etc.) That said, employees teleworking eight hours or less may be at a decreased risk of experiencing depression, while those working extended telework hours may experience depressive symptoms at the hand of social isolation and reduced social support. Similar findings have been reported in relation to psychological well-being. In some cases, the reductions in social support associated with telework usage leads to lower levels of job engagement. On the other hand, those utilizing telework to reduce job demands see positive effects for employee well-being.

Finally, we identified a number of social and work-related outcomes associated with engaging in telework. The flexibility and control associated with telework may help bolster work–life balance and reduce work–family conflict. However, reductions in spatial and temporal boundaries between work and home may increase the likelihood of family-to-work conflict and increase stressful experiences for workers inside of the home. These outcomes seem to largely relate to one’s boundary management style. Considering work-related outcomes, teleworking employees often report increased job satisfaction, although these reports slightly taper or plateau when employees begin to work extensive hours via telework. In addition, telework is associated with reduced absenteeism, but this relationship may be due to a health trade-off in which employees more frequently continue working when sick away from the office out of concern for losing their ability to telework in the future. Nonetheless, the bulk of these findings indicates the importance of building a multi-disciplinary understanding of the relationship between telework and worker health and have important implications for the design of teleworking arrangements now and in the future.

### 9.1. Recommendations for Policy and Organizational Practice

One of the challenges with telework is that there are no federal regulations about the use or implementation of telework. Instead, work arrangements are often left up to an individual employer, or in some cases, one’s supervisor [105]. First, we acknowledge and understand that not all jobs are conducive to telework. However, experiences during the COVID-19 pandemic have demonstrated that telework may be more feasible for some jobs than previously thought. Our review highlights the importance of having a clear, well-communicated and inclusive telework policy.

#### 9.1.1. Telework Policy

When considering telework policy, organizations are ultimately tasked with balancing the flexibility and formality. Overall, informal, or as-needed, telework arrangements are more likely to increase worker perceptions of autonomy and flexibility; however, formal arrangements can benefit employee performance as well as perceptions of job control when presented as a flexible support mechanism [47]. Regardless of the formality of an organization’s telework policy, supervisors and human resource personnel should ensure clear and consistent criteria for establishing who is eligible to telework, the location of telework, as well as when and how often an employee may telework. Furthermore, practitioners should be careful to not provide telework as a replacement for formal family-supportive or other support policies such as paid sick time, as employees may be more likely to experience work–family conflict and presenteeism, subsequently impacting employee health, well-being, and performance.

#### 9.1.2. How Often, When, and Where to Telework

Research to date indicates that the optimal time spent teleworking is approximately 40% of one’s overall working hours, equating to two 8-h workdays under the conventional 40-h work week. However, organizational leaders and managers should recognize that working more time via telework does not lead to detriment in worker health or performance, but likely other factors prevent further gains in worker satisfaction. Furthermore, leaders should consider when an employee chooses or is scheduled to telework. Workers have been shown to experience adverse health effects when telework is used as a mechanism to catch up on work after hours or over the weekend. Telework is most beneficial for employee health and well-being when provided as a flexible support and not in lieu of formal support such as paid time-off. Supervisors should remain aware of their employees’ workload, work hours, and teleworking behaviors in order to mitigate instances of employees teleworking after hours to “make up” for missed work due to scheduled work time off, such as vacation, paid time off, or sick time.

Workers are most likely to experience the beneficial outcomes for health and well-being when teleworking from a location where they have the greatest level of control of the work environment. For instance, workers may be able to better control the levels of lighting, noise, and temperature within their homes than a library or co-working space. However, certain aspects of working from home such as separating family and nonwork responsibilities or interruptions from one’s work domain may be more difficult. Therefore, workers will likely benefit from choosing a consistent space within their homes to perform their work where non-work activities do not take place and they can best separate their work and home domains. For instance, when possible, workers may choose to work from a separate room where they may close the door at the end of their workday, or work from a table in a shared room specifically designated for work tasks.

#### 9.1.3. Managing Boundaries When Teleworking

Telework reduces the likelihood for work interfering with employees’ nonwork domains [46], but also provides a greater opportunity for nonwork interference during one’s workday. Along with ensuring a separate physical space within one’s home where work is performed, workers might also set clear expectations for both work and nonwork communications when teleworking. For instance, in order to circumvent employees working after hours when teleworking, both supervisors and employees should discuss the expectations for responding to work-related communication. Additionally, workers should communicate their expectations to family and friends when teleworking in order to mitigate the likelihood for non-work interruptions.

#### 9.1.4. Training, Technology, and Ergonomic Support

As organizations continue to implement telework policies within their workplaces, special attention should be given to the educational, technological, and ergonomic resources provided to employees. Practitioners should strive to implement telework training and provide sufficient ICT to the greatest extent possible, so that employees feel that they have adequate resources to meet the demands they experience when teleworking and also reduce stress invoked from insufficient technology. One option which organizations might consider is the quality of wireless internet employees have within their homes, and the extent to which their organization can help in ensuring that their wireless connections are adequate for the work they are expected to perform. Organizations may also consider and provide workers with a practical stipend to purchase, or otherwise provide, ergonomic furnishings for their in-home workspaces, such as an ergonomic work chair which they may be accustomed to when working from their organizational setting. Another recommendation is to provide professional ergonomic assessments for teleworkers. Ensuring employees have a healthily designed workspace in their homes will likely reduce the development of musculoskeletal and pain symptoms associated with poorly designed workstations.

#### 9.1.5. Retaining Autonomy

The onset of the 2019 coronavirus pandemic has led to extreme shifts in the way that work is organized and performed. For instance, in many cases, employees have no longer received the choice to telework and were mandated to work from home by either state or organizational guidelines. The removal of the choice of where work is performed may alter what we already know about the importance of autonomy for occupational health. By removing this choice, such that many workers are involved in mandatory telework, there is the potential for increased stress and adverse effects related to employee health and well-being [88]. Under these circumstances, practitioners and leaders will need to identify additional job resources, such as social support or flexibility among workers’ schedules, to alleviate the potential for reduced perceptions of autonomy as well as the unique job demands experienced when teleworking by many, such as working in the presence of family or partners.

#### 9.1.6. Maintaining Social Connections

It is likely that many employees are now teleworking beyond the optimal extent of telework (i.e., 40%), and although research has largely unearthed the “telework paradox”, some employees may experience loneliness when teleworking at high intensities. Thus, management and supervisors should aim to provide as many face-to-face interactions with teleworking employees as possible, and especially with new remote employees where face-to-face interactions are crucial for healthy socialization during their first 90 days [106]. Online web conferencing platforms (e.g., Zoom, Microsoft Teams) may help supervisors meet these needs, and informal channels such as Slack may benefit employees who value informal and unscheduled interactions with colleagues.

### 9.2. Future Research

#### 9.2.1. New Normal of Teleworking

In the coming years, it is likely we see an increase in the number of full-time regular teleworkers [4]. Occupations that have been traditionally confined to the working office due to organizational norms are now being practiced from home via computer technologies, largely as a result of the COVID-19 pandemic. As many employees may now be working more hours by telework, now is the time for researchers to expand what we know about the extent of telework, as previous studies have rarely investigated full-time or almost full-time teleworking employees.

Furthermore, working adults with children are now more likely to be attending to childcare responsibilities as many k-12 schools closed or moved to an online format during the COVID-19 pandemic and some schools have retained these practices within high-risk populations. Thus, our current understanding of the impact of telework on work–life and work–family outcomes may change as a result of the pandemic and new teleworking norms. In a similar vein, a greater number of married and co-living partnerships may both be working from within the home both during and after the pandemic, and studies prior to the COVID-19 pandemic have yet to unpack the intricacies of co-working partners. Future research will need to consider how the changing organization of work and family roles while teleworking impacts employee health and well-being, particularly over the long term.

Future research should also aim to investigate ways in which workers’ socioeconomic status relates to their teleworking experiences and outcomes. For instance, during the COVID-19 pandemic, many workers transitioned to remote work without necessary or familiar ergonomic and technological resources. Meanwhile, some workers may not have had adequate financial resources for purchasing their own ergonomic workstations or updating their in-home technology. For example, less than half of teleworkers responding to a 2020 global work-from-home survey reported having ergonomic supports such as a sit–stand desk, dual or wide-screen monitors, or ergonomic chair, despite over half of respondents indicating having these supports when working from their physical organization [107]. Additionally, some workers may live in environments that have excessive noise contributing to frequent disruptions when teleworking (e.g., due to construction, traffic, etc.), or where they are unable to control the micro-climate of their physical location (e.g., no central air-conditioning).

Thus, future research might also consider socio-economic status as a moderator within the relationship between telework and worker health and well-being. Although we did not include socioeconomic status as a moderator in our conceptual model, the moderating role of socioeconomic status seems plausible. Similarly, future research might consider the potential moderating role of other variables presented in our conceptual model of the relationship between telework and worker health and well-being. While we did not identify empirical articles discussing a moderating role of physical activity or sleep within the teleworking context, it is possible that these factors may alter outcomes when considering teleworker health and well-being, and future research should investigate these issues.

#### 9.2.2. Underrepresented Groups

People with disabilities and chronic health conditions (CHC) experience a disproportionate burden of unemployment, and the COVID-19 pandemic has begun to exacerbate ability-based differences in employment, with the employment rates of individuals with disability decreasing at a greater rate (11.2%) than those without disability (6.7%) amid the beginning of the pandemic. The Americans with Disabilities Act of 1990 promoted the use of telework when considering the hiring and retention of individuals with disability and CHCs. However, organizations are not required to provide telework as an accommodation or to create a more inclusive work environment unless the nature of the work is deemed acceptable for telework and allows for workers to meet the essential functions of their job [26]. Many courts have ruled against telework as a reasonable accommodation, as teleworking requires an employee to be absent from their central workplace and attendance has often been considered a necessary component to one’s job [108]. However, as the prevalence of teleworking employees in general continues to rise as a result of the COVID-19 pandemic [4], telework accommodations for workers with disability may become more likely. For instance, in 2020, the World Health Organization endorsed telework for workers with disability during the pandemic in order to reduce concerns of exposure to COVID-19 [109].

However, there is a dearth of literature investigating the utility of telework as an accommodation practice for these workers. In general, employees recognize telework as a means to alleviate work interference with family and also manage pain or fatigue not associated with disability [110]. Meanwhile, some workers are utilizing telework as an accommodation practice through their employer [111]. However, only half of employees report satisfaction with their telework accommodation, despite a majority (76%) reporting that teleworking was beneficial in completing their work tasks. These findings suggest that there may be effects related to health and well-being influencing the teleworking experience of workers managing disability or chronic conditions.

A recent study provides preliminary insight into the beneficial mechanisms of telework for supporting workers with disability or chronic health conditions (CHCs) [112]. In this study, participants with disability and CHCs who completed daily measures of job control, flexibility, work ability, and well-being indicated significantly higher levels of job control when teleworking. Increased reports of job-control among participants, in turn, were associated with higher-levels of perceived work ability and well-being. However, this study was conducted during the height of the COVID-19 pandemic when most participants were teleworking at very high-intensities. Future work is needed to determine how telework acts as an accommodation practice among various intensities of telework (e.g., on an as-needed basis versus several days of scheduled telework a week).

Finally, future work ought to extend research to include pregnant women or workers responsible for eldercare. In Australia, the Fair Work Act extends the right to employees with caring responsibilities, a disability, or caring for a family or household member experiencing violence to request a flexible work arrangement in order to effectively navigate work and personal needs [26]. In the United States, the courts may impose and direct an organization to provide telework as an accommodation for pregnant workers [55]. Empirical studies evaluating the utility of telework as a flexible work arrangement under these conditions is needed to ensure that researchers, organizational leaders, managers, and policy makers understand the components of telework which best meet the needs of these special groups.

#### 9.2.3. Methodology

Although interest in designing and implementing telework studies has surged in recent years [2] and even more recently due to COVID-19, much of our current understanding of telework outcomes is constrained by limited methodology. For instance, the majority of the studies included in this review utilized a cross-sectional survey. Although there are ways to optimize the utilization of cross-sectional methods [113], cross-sectional designs are not conducive to investigating change over time. Only a handful of studies included within the current review used longitudinal data. In one example, Vega et al. [39] utilized a longitudinal design in which participants completed daily surveys for five consecutive workdays in order to evaluate changes in job satisfaction, creativity, and performance. Longitudinal designs of this nature allow us to understand the effects of telework from a dynamic and within-person perspective. For instance, future work may look to further evaluate changes in employee health and well-being as a result of working standard versus non-standard working hours.

Another methodological concern constraining the field’s ability to generalize results across studies is the duration of time an organization has implemented teleworking programs. For instance, it is likely that workers within an organization with a well-established telework program will have differing teleworking experiences than those within an organization newly implementing such programs. Given the number of organizations with newly developed teleworking policies in light of the COVID-19 pandemic, future research might draw upon or develop new theories to understand why differences are seen as a result of how long an organization has offered telework.

The types of data and measurement tools used within telework and health studies is also limited. Ambulatory methods which mitigate disruption and also collect objective health data [114], such as wearable blood pressure and heart rate monitors, may be useful in directing researchers to understand stress and other physiological functions as a result of extended telework usage. For example, there is an abundance of non-invasive health tools for measuring sleep, endocrine, and cardiovascular activity. Future researchers might consider the usefulness and benefits of these measurement tools for understanding teleworker health and well-being.

Furthermore, when surveys are used to investigate relationships between telework and worker health and well-being, investigators ought to consider the extent to which measures developed for application within the office or traditional work environment also apply among various telework settings. For instance, researchers have begun to consider the applicability of work–family measures which conceptualize work and home as distinct geographic locations [115]. Additionally, when considering employee organizational citizenship behaviors (OCBs, i.e., positive or helping behaviors directed at individuals or the organization which benefit organizational goals), prior research reports an equivocal relationship between telework and OCBs, potentially due to measurement issues [116]. Since OCBs are typically bound to one’s physical work environment, investigators will have to determine if, and how, these behaviors change within a telework context, and subsequently develop and validate appropriate measures. Thus, researchers should consider how their chosen measures lend to the virtual work environment and ensure appropriateness using statistical analyses such as confirmatory factor analyses and testing measurement equivalence between telework and non-teleworking groups.

Finally, as the ways and frequencies by which employees continue to telework increase, researchers must also consider and come to a consensus regarding how to measure telework itself. In a recent meta-analysis [46], authors gathered 22 empirical articles investigating the relationship between telework and bi-directional measures of work–family conflict. Among these studies, approximately half measured telework dichotomously (e.g., teleworkers compared to non-teleworkers), while the remaining studies measured telework continuously (e.g., days per week, hours per week, the extent of telework, etc.) Beckel et al.’s results found that measurement differences moderated the relationship between telework and work–family conflict, such that studies including a dichotomous measure of telework exhibited a stronger, negative relationship between telework and work–family conflict. However, dichotomizing variables that can be measured on a continuum may result in a loss of information and can introduce error [117]. Thus, future research should better utilize continuous measures of telework, such as the extent of telework, especially as workers utilize their options to telework differently, ranging from on an as-needed basis to full-time teleworking.

Researchers might also consider the inclusion of employer-provided health risk assessment and health care utilization data. For instance, health care utilization data might help direct researchers in understanding the assumed health trade-off discussed by Mann and Holdsworth [19] wherein employees under-utilize their access to health care and in turn continue teleworking when sick (i.e., presenteeism) to maintain their teleworking privilege. These data might also be helpful in determining whether telework is a beneficial accommodation practice for those managing chronic illness or disability when compared with other forms of cross-sectional data. For instance, do workers with a disability or CHCs use their telework accommodations in order to manage higher rates of healthcare appointments when compared to workers without these conditions? Nonetheless, as there is much further to go in the investigation between telework and worker health, broadening both our methods and measurement tools will be an integral component to also advancing our understanding about the outcomes associated with telework.

### 9.3. Limitations

The primary limitation of this article is that we performed a narrative review of the literature rather than an empirical meta-analysis. However, our review summarizes a broad array of factors related to antecedents and outcomes of telework that would be challenging to incorporate in a meta-analysis, especially considering the various ways in which telework has been measured across studies.

Furthermore, because much of the telework literature generated before the year 2000 is included in other reviews [1,2,5], we only included earlier articles when they were most relevant. We also chose to omit discussion relating to telework and government policy, regulations, worker’s compensation, case law, and organizational policy. Prior reviews and articles, such as Blount [26], Montreuil and Lippel [55], and Allen et al. [2] provide preliminary discussions of these topics.

Given the rapid increase in telework prevalence since the onset of the COVID-19, pandemic, it is likely in the years to follow, there will be an increase in empirical studies investigating the impact of telework on employee and organizational outcomes. As we learn more about the effects of telework on worker health and well-being, we encourage researchers to further expand upon the conceptual framework presented in this review.

### 9.4. Conclusions

This article advances the occupational health and public health literature by reviewing empirical studies to explain the relationship between telework and worker health and well-being. There are a variety of components which contribute to our understanding of the benefits and consequences of telework for worker health and well-being. Individual worker and job characteristics, the social context of work, and the organization of personnel and technological systems help us to understand the variety of health outcomes presented within our review. We provided a conceptual model using both the job demands-resources and macroergonomic systems approach to illustrate the multidisciplinary components which contribute to the telework and occupational health relationship. Thus, as working dynamics change throughout the progression, and hopefully cessation, of the COVID-19 pandemic, both researchers and practitioners will need to prepare for actions to meet the changing needs of employees. We hope this review will help guide the development and implementation of federal regulations, organizational policies, and procedures to support telework practices that support worker health and well-being. We also hope this article provides a foundation and organizing framework to guide future research related to telework and worker health and well-being.

## Figures and Tables

**Figure 1 ijerph-19-03879-f001:**
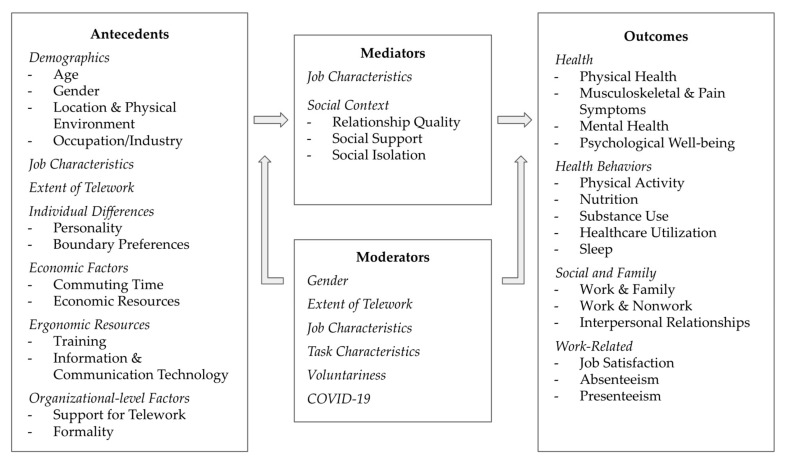
Conceptual model of telework and worker health and well-being.

## Data Availability

Not applicable.

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
