# Peer review of "Telework and Worker Health and Well-Being: A Review and Recommendations for Research and Practice"

_ijerph, 2022, doi:10.3390/ijerph19073879_

Round 1
Reviewer 1 Report
The paper summarises well-being research relevant to teleworking. This analysis is presented in the form of a conceptual model, and emphasis is placed on the changes to telework resulting from CV19. Overall, the usefulness of this item is high, and the readability excellent. The inclusion of a section on underepresented users is particularly welcome. I have only a few minor concerns.
1) The conceptual model, while sensible, is not well grounded in the paper. Can the authors at least explain whether the model comes from the literature review, or was already extant or derived from a pre-existing source. How does it relate to the theoretical background. It might be more sensible to include Section 4 within Section 3 (eg as Section 3.1).
2) I would question the definition (or lack of definition) of ergonomics, which is very specific to physical design of workstation and the usability of ICT. The IEA or CIEHF definitions of ergonomics would encompass many aspects of your paper. I dont think what the authors are saying is incorrect, within the section but I would prefer the authors to preface this section with a definition of what is in scope.
Reviewer 2 Report
Thanks for the opportunity to review this manuscript.
During the pandemic of C0vid-19, the most significant point is how to keep healthy. For safety, people need to maintain social distance. Therefore, telework become an alternative way to balance work and safe. Does telecommuting affect people’s health? This review gives an important viewpoint.
My comments are as followed.
Firstly, a structured abstract is needed. I would like to recommend using the following structure to present the key information about this study, including objective; background; methods; discussion; conclusion in the Abstract.
Secondly, is there any empirical studies focused on telework and worker health and well-being? It is encouraged to briefly introduce the impacts of telework on workers’ health and well-being in the Introduction. Thirdly, that will be better to add the inclusion/exclusion criteria into the Methods. This can make clear what is this review focus on and the relevance of included studies.
Reviewer 3 Report
This is a very interesting review which outlines the state of the literature evidence regarding the relationship between telework and worker health and wellbeing. The review is very well written and organized and discusses a very broad and variable field. Below are some considerations for the authors:
- Can the authors provide their reasoning for using Google Scholar and PRIMO search engines compared to MEDLINE and other more established electronic search databases with larger academic collections and detailed search indexing algorithms? Google Scholar has also been shown to have low efficacy in returning grey literature yields.
- Can the authors specify if there were any language or regional limits set for the search strategy?
- The authors should be commended for a very thorough and appropriate conceptual model for their review. For me, missing was the inclusion of a measure of a person’s socio-economic situation in the model, which I see as potentially as a moderating factor. For example, some people have been forced into telework without their employer considering whether they have adequate space/comfortable chairs and desks at home. Some workers might be sharing their working space with their family members, which might impact their psychological wellbeing and interpersonal relationships at home. Telework would require a stable internet connection and a capable computer (there was some discussion of this in section 5.6.2) – but some workers might not have these available to them or are sharing these facilities with family members also working or studying at home, which in turn might have effects on the different outcomes listed in the conceptual model. Discussion of these points and others like them might also be relevant to note in sections 5.1.3 (location and physical environment) and 5.4.2 (boundary preferences).
- The review mentions foregoing any consideration of telework in relaton to government polic, regulations, and organizational policy. Nonetheless, these factors underpin the links between telework and health outcomes and should be identified in relation to the other constructs within the paper’s conceptual model.
- Health behaviors are listed as outcomes in the conceptual model but some could be moderating/mediating factors. For example, there can be gender differences in the time availability for physical activity, which in turn can influence gender differences in physical/psychological wellbeing and musculoskeletal health.
